# Differentiable Cell Complexes Discover Interpretable Metabolic Connectivity Subtypes in Alzheimer's Disease

**Oankar R. Patil**[*1] (iD)                                    OPATIL31@GATECH.EDU
[1] *Department of Electrical and Computer Engineering, Georgia Institute of Technology*
**Keenan Hom**[*2]                                             KHOM3@GATECH.EDU
[2] *Department of Computational Science and Engineering, Georgia Institute of Technology*
**Daniel Drane**[3]                                            DDRANE@EMORY.EDU
[3] *Department of Neurology, Emory University*
**May D. Wang** [4]                                            MAYWANG@GATECH.EDU
[4] *Wallace H. Coulter Department of Biomedical Engineering, Georgia Institute of Technology*

## Abstract

Data-driven subtyping of Alzheimer's disease (AD) using conditional variational autoencoders (cVAEs) has identified metabolic subtypes from FDG-PET, but existing approaches provide no insight into which inter-regional metabolic relationships define each subtype. We introduce a parcellated cVAE with a Differentiable Cell Complex Module (DCM) that learns higher-order topology over atlas-parcellated brain regions, enabling simultaneous subtype discovery and interpretable connectivity mapping. Applied to 716 AD subjects from ADNI, our model identifies two severity-matched subtypes with anti-correlated connectivity ($r=-0.81$), distinct cognitive profiles ($p<0.001$), and differential CSF tau ($p=0.0008$, corrected), corresponding to posterior-cortical and limbic AD variants.

**Keywords:** Topological Deep Learning, FDG-PET, Alzheimer's Disease

## 1. Introduction

Alzheimer's disease (AD) exhibits substantial heterogeneity in patterns of neurodegeneration, with established subtypes including posterior-cortical, limbic-predominant, and typical variants that differ in cognitive profiles and biomarker trajectories (Ferreira et al., 2017). Data-driven approaches using conditional variational autoencoders (cVAEs) on FDG-PET have successfully identified metabolic subtypes by clustering in a latent space conditioned on disease severity. However, these models treat the brain as a flat feature vector, providing cluster assignments without revealing which inter-regional metabolic relationships characterize each subtype (Ryoo et al., 2024). Separately, topological deep learning (TDL) has introduced cell complex neural networks that perform message passing over nodes, edges, and higher-order cells, capturing multi-way relational structure beyond pairwise graphs (Hajij et al., 2020) (Bodnar, 2023). Yet TDL methods have not been integrated with generative models for unsupervised disease subtyping, nor applied to brain network analysis where higher-order regional interactions have clear neurobiological significance. We propose a parcellated cVAE with a Differentiable Cell Complex Module (DCM) that learns task-specific topology over AAL3 atlas regions from FDG-PET data, enabling simultaneous subtype discovery and interpretable metabolic connectivity mapping. Applied to 716 AD subjects

---

[*] Contributed equally

from ADNI, our model identifies two severity-matched subtypes with anti-correlated spatial connectivity, distinct cognitive profiles, and differential CSF tau pathology, corresponding to established posterior-cortical and limbic-predominant AD variants.

## 2. Methods

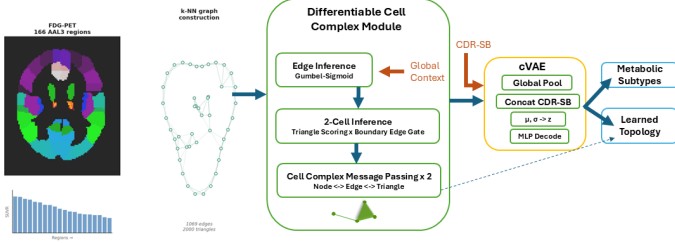

Figure 1: Parcellated DCM-cVAE architecture.

### 2.1. Data and Parcellation

We used FDG-PET scans from 716 AD subjects in the Alzheimer's Disease Neuroimaging Initiative (ADNI) (Petersen et al., 2010). Each scan was spatially normalized to MNI space and parcellated into 166 regions using the AAL3 atlas (Rolls et al., 2020). Regional features were computed as mean SUVR per region and z-scored across subjects.

### 2.2. Differentiable Cell Complex Module (DCM)

DCM was introduced for latent topology inference in classification tasks (Battiloro et al., 2023). We extend this framework to conditional generative models for unsupervised disease subtyping. We construct a candidate graph by connecting each region to its k=10 nearest spatial neighbors (1069 undirected edges, 2000 candidate triangles). Regional features are projected to node embeddings with positional encoding from atlas centroids. The Edge Inference Module scores each candidate edge using projected node pair features and a global context vector (mean-pooled across all nodes), producing per-sample edge probabilities via Gumbel-Sigmoid relaxation. This global context enables subtype-specific edge activation, with different input patterns producing different learned topologies. The 2-Cell Inference Module similarly scores candidate triangles, gated by boundary edge probabilities to enforce topological consistency. Two layers of Cell Complex Message Passing propagate information across nodes, edges, and 2-cells, weighted by the learned probabilities.

### 2.3. Conditional VAE

Pooled node and edge features are concatenated with CDR-SB as condition, then projected to latent mean and variance. An MLP decoder reconstructs regional features from the latent code. The loss combines MSE reconstruction, MMD divergence (Gretton et al., 2012), and topology sparsity term targeting 20% edge density. Subtypes are identified by applying k-means clustering to latent mean vectors, with k selected via silhouette score/elbow plot.

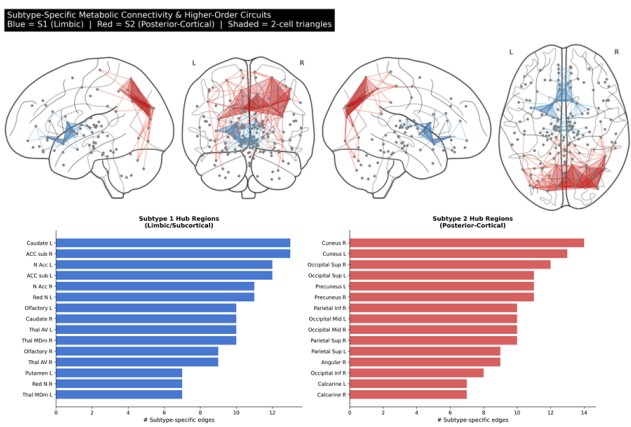

Figure 2: Extracted topology of severity-matched metabolic subtype connectivity.

## 3. Results and Conclusion

Applied to 716 AD subjects from ADNI, our model identifies three subtypes: a milder-severity group (n=288) and two severity-matched higher-order topology subtypes (n=242, n=185; CDR-SB p=0.309) with anti-correlated learned connectivity (r=-0.81). The posterior-cortical subtype (S2) exhibits parietal-occipital metabolic connections with significantly worse executive and visuospatial function (p<0.001) and elevated CSF tau (p<0.001, surviving age and sex correction at p=0.0008), while the limbic subtype (S1) shows olfactory-subcortical connections with preserved cortical function (Table 1, Figure 2). A standard MLP cVAE on same parcellated features failed to differentiate on cognitive domains/tau pathology.

Table 1: Clinical profiles of two severity-matched AD metabolic subtypes

| Variable | S1 | S2 | $p$ |
|---|---|---|---|
| *Severity* | | | |
| CDR-SB | 5.88 (2.62) | 5.64 (2.63) | 0.309 |
| MMSE | 21.65 (4.29) | 21.18 (4.17) | 0.171 |
| *Cognitive domains* | | | |
| Executive | −0.44 (0.73) | −0.89 (0.70) | <0.001 |
| Visuospatial | −0.12 (0.75) | −0.57 (0.91) | <0.001 |
| Language | −0.34 (0.69) | −0.49 (0.63) | 0.007 |
| Memory | −1.00 (0.52) | −0.96 (0.50) | 0.564 |
| *CSF biomarkers* | | | |
| P-tau | 30.4 (13.1) | 37.5 (14.9) | <0.001 |
| Total tau | 318.0 (116.9) | 388.6 (148.4) | <0.001 |
| *Learned topology (S1 vs S2)* | | | |
| Edge correlation | $r = -0.81$ | | |
| Divergent edges | 334 / 1069 | | |

## Acknowledgments

Partnership for an Advanced Computing Environment (PACE) at the Georgia Institute of Technology, Atlanta, Georgia, USA (RRID:SCR_027619). Data used were obtained from ADNI (adni.loni.usc.edu).

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
