# OpenReview forum: "Differentiable Cell Complexes Discover Interpretable Metabolic Connectivity Subtypes in Alzheimer's Disease"
_MIDL.io/2026/Short_Papers — MIDL 2026 - Short Papers Poster_

### Official Review · Reviewer_yM1W · 2026-05-03
**Promising combination of topological deep learning and AD subtyping.**

**Rating:** 5
**Confidence:** 4

**Review:**

See strengths and weaknesses.

**Summary:**

The paper addresses an interpretability gap in data-driven AD subtyping that
prior cVAE-based clustering of FDG-PET recovers
metabolic subtypes but does not say which inter-regional metabolic
relationships define each subtype. The proposed approach is a parcellated
cVAE conditioned on CDR-SB, that infers per-sample edges and
2-cells over a graph using Gumbel-Sigmoid
relaxation. After two layers of cell-complex message passing, pooled
features are projected to a latent space; subtypes are identified by
k-means on latent means with k chosen by silhouette/elbow. The model
is trained on 716 ADNI subjects. The reported result is three clusters
(one milder severity, two severity-matched), where the two severity-
matched groups (n=242, n=185, CDR-SB p=0.309) have anti-correlated
learned connectivity (r=-0.81), differ in executive and visuospatial
cognition (p<0.001), and differ in p-tau and total tau (corrected
p=0.0008). These are claimed to correspond to posterior-cortical and
limbic-predominant AD variants from the literature.

**Strengths:**

1. Extending DCM (originally introduced for
  classification with latent topology inference) to a conditional
  generative model for unsupervised subtyping is a novel contribution. It is very intriguing and bears potential to other related tasks.

2. The conditioning on CDR-SB is a reasonable choice that allows
  severity-matched comparisons. The authors clearly have thought about the problem because metabolic subtypes
  are easily confounded with disease stage.

3. Relatively large cohort for an FDG-PET subtyping study (n=716 from
  ADNI), with a clean parcellation pipeline (MNI, AAL3, z-scored SUVR).

4. The reported clinical separation is internally consistent: anti-
  correlated edge sets, differential cognitive domain scores, and
  differential CSF tau biomarkers all point in the same direction.

**Weaknesses:**

1. There is some inconsistency in the number of subtypes between abstract and results.
  The abstract claims "two severity-matched subtypes," but the Results
  state "our model identifies three subtypes: a milder-severity group
  (n=288) and two severity-matched higher-order topology subtypes." Is it really two or three, I was confused.

2. Another consistency issue. The "1334 / 1069 divergent edges" entry in Table 1 appears to be a
  typo or unexplained metric? The candidate graph has 1069 edges and 1334 cannot be a subset. Unless I missed something.

3. This is a little bit picky for a 3-page paper, but I am pointing this out anyways (for future work). I found some ablation study necessary. Without comparing against (a) the same parcellated cVAE
  without DCM, (b) a vanilla cVAE on flat features, or
  (c) a fixed-graph GNN cVAE, it is impossible to tell whether DCM
  actually contributes anything beyond cVAE clustering. The whole pitch
  hinges on DCM being necessary, but no experiment isolates its effect.

**Justification Of Rating:**

This is a genuinely interesting short paper that combines topological deep learning and Alzheimer's disease subtyping in a non-trivial way. Extending the a cell complex module from its original latent-topology classification setting into a cvae is very intriguing. The empirical results are internally consistent across connectivity, cognitive etc. There are no major issues as a 3-page paper, but I listed some typo/inconsistencies in the results (this should be fixed), and I encourage the authors to include ablation studies should they extend paper to a full length or journal version.

Despite these issues, the novelty of the methodological framing and the coherence of the clinical findings outweigh the limitations for a short-paper submission, and I recommend acceptance.

---

### Decision · Program_Chairs · 2026-05-08

Accept (Poster)